# Relative Contribution of Citizen Science, Museum Data and Publications in Delineating the Distribution of the Stag Beetle in Spain

**DOI:** 10.3390/insects12030202

**Published:** 2021-02-27

**Authors:** Marcos Méndez, Fernando Cortés-Fossati

**Affiliations:** Area of Biodiversity and Conservation, Universidad Rey Juan Carlos, c/ Tulipán s/n., Móstoles, E-28933 Madrid, Spain; fernando.cfossati@urjc.es

**Keywords:** crisis of faunistic entomology, historical data, insect conservation, *Lucanus cervus*, Wallacean shortfall

## Abstract

**Simple Summary:**

Conservation of insects requires a reliable knowledge of their distribution. Such knowledge is hard to obtain in many cases, due to lack of human power and funding for extensive surveys. Three ways out of this problem have been suggested: (1) data already available in museum collections, (2) data already available in the entomological literature and (3) use of citizen science projects as a cheap, efficient way to survey extensive territories. We assessed the contribution of each of these sources of information in delineating the Spanish distribution of the European stag beetle. Although citizen science quickly contributed more grid cells than the other sources, some grid cells were uniquely contributed by museum and publication data. Thus, the three sources of information need to be combined when targeting endangered species in a broad, heterogenous, sparsely populated territory such as Spain.

**Abstract:**

Reliable distribution maps are in the basis of insect conservation, but detailed chorological information is lacking for many insects of conservation concern (the Wallacean shortfall). Museum collections, entomological publications and citizen science projects can contribute to solve this Wallacean shortfall. Their relative contribution to the knowledge on the distribution of threatened insects has been scarcely explored, but it is important given that each of these three sources of information has its own biases and costs. Here we explore the contribution of museum data, entomological publications and citizen science in delineating the distribution of the European stag beetle in Spain. Citizen science contributed the highest number of records and grid cells occupied, as well as the highest number of grid cells not contributed by any other information source (unique grid cells). Nevertheless, both museum data and publications contributed almost 25% of all unique grid cells. Furthermore, the relative contribution of each source of information differed in importance among Spanish provinces. Given the pros and cons of museum data, publications and citizen science, we advise their combined use in cases, such as the European stag beetle in Spain, in which a broad, heterogeneous, sparsely populated territory has to be prospected.

## 1. Introduction

A first step in assessing the conservation status of any insect is a reliable distribution map. Distribution maps lay at the basis of three conservation tools: (1) prioritization of areas to be protected [1,2,3], (2) estimation of extinction risk [4,5] and (3) niche and habitat models [2,6]. However, distribution ranges of many insects are deficiently known (the Wallacean shortfall [7,8]), including those of conservation concern. For example, IUCN red lists of European saproxylic beetles rated 24–49% of the species assessed as Data Deficient [9,10,11]. Among the reasons of this shortfall are its logistic challenge [12], combined with underfunding and lack of researcher power [8].

A first way to solve the Wallacean shortfall for insects is the regular publication of chorological and faunistic papers in entomological journals. For example, the number of faunistic publications for Iberian butterflies has exponentially increased over time [12]. However, faunistic publications are increasingly seen as scientific output of low academic excellence and discouraged among professional scientists [13]. As a consequence, the number of professional and amateur taxonomists is declining in some developed countries [14] and many national scientific journals face an uncertain future [15].

Two additional solutions to the Wallacean shortfall, of direct use in conservation, were offered by [8]: compilation of data in public repositories, such as museums, and citizen science. Museums are important sources of data for conservation [16,17,18,19] and biological collections have increasingly contributed to ecological and environmental studies [20], including mapping [21]. Two potential limitations of museums are their geographical bias [20] and that underfunding of natural history research [22] is decreasing the ability to sustain the growth of natural history collections [23]. Citizen science has also gained momentum as a monitoring method [24], including entomology [25]. Among its advantages are its ability to cover large geographic areas [26] and mobilize a high number of volunteers [27]. Thus, its potential largely exceeds from the efforts of a limited number of researchers. Drawbacks of citizen science include lack of taxonomical expertise, which can result in unreliable identifications, as well as a potential bias towards urban and suburban habitats [28,29] and costs of coordination, including keeping regular feedback and motivation [30].

Researchers or managers wishing to design a successful project for mapping the distribution of a threatened insect should consider the pros and cons of different strategies of data acquisition. Although the decision tree is potentially complex [30], here we focus on the contribution of citizen science compared to publications and museum data. The relative contribution of citizen science, publications and museum data has been assessed in relation to the Linnean shortfall, i.e., the lack of knowledge on species richness [31,32]. We are unaware of any similar attempt for the Wallacean shortfall.

The European stag beetle, *Lucanus cervus* Linnaeus, 1758, the largest beetle in Europe, is included in the Bern Convention, the Habitat Directive and the annex II of Natura 2000 [33]. Its conspicuous size and shape make it a very popular beetle among entomologists and the general public, with sustained attention in the entomological literature [33] and representation in iconography through the European history [34]. In addition, it is easy to recognize by the general public, thus making it a suitable candidate to citizen science programs. In fact, its range size has been delineated using citizen science programs in several European countries (France: [35]; Italy: [36]; Portugal: [37]; Spain: [38]; UK: [28,39,40]). Citizen science can be efficient in delineating the distribution of the stag beetle compared to historical data [28,36]. Although mismatches were found between recent and historical records [28,36], no comprehensive assessment has been made of the relative contribution of different sources of data to the distribution of the stag beetle. In particular, it is relevant to ascertain whether citizen science adds new grid cells, thus covering sites that have not been previously explored by traditional entomological prospection, or fails to confirm the presence in sites previously reported by traditional entomological prospection.

The main goal of this study was to assess the relative contribution of citizen science, museum data and publications in delineating the distribution of the European stag beetle in Spain. This goal was addressed through three specific objectives. First, the extent to which each of the data sources—citizen science, museums and faunistic publications—provided unique versus redundant information was assessed. Second, the accumulated contribution of each data source through time was compared. Third, to highlight geographical differences in the contribution of each data source, their relative contribution was assessed at a provincial level.

## 2. Materials and Methods

### 2.1. Database Compilation

A database of stag beetle records in Spain was built, starting in 1994, based on three sources: published literature, museum data and citizen science. Published literature (hereafter “publications”) included entomological papers, books, technical reports, popular articles and atlases reporting records of stag beetle. A total of 76 publications, published between 1784 and 2019, were included in the database. Museum data (hereafter “museum”) were collected from 28 public collections in 17 faculties of 15 Spanish universities, six Spanish museums, two international museums, two Spanish scientific societies and one Spanish research centre. Most of these collections were visited once, between 1994 and 2006. Only one collection was revisited six years, between 1996 and 2009. In all cases, all published records were transferred to the “publications” section. Citizen science data were provided personally or by e-mail by more than 525 people, from 1994 to 2019, in response to direct requests or data requests through national entomological journals, nature magazines, online subscription lists and the homepage of the Grupo de Trabajo sobre Lucanidae Ibéricos, and consisted of unpublished records sent by naturalists and general public. These data include unpublished data from private entomological collections. Data from citizen science platforms were less than 0.3% of all records. Large virtual repositories such as Biodiversidad Virtual, observado.org or iNaturalist were not included due to lack of workforce to assess the accuracy of the records. This database has been the basis for the periodical assessments of the status of the European stag beetle in Spain in response to the mandate of the EU. Preliminary assessment of its coverage by means of habitat models (unpublished data) indicate that current data do not miss any major areas of Spain in which the European stag beetle could be present. Major empty areas are due to unsuitable environmental conditions and not to lack of prospection. Distribution gaps remain to be filled but only in provinces in which this beetle has already been reported and usually new cell grids reported annually are close to already occupied localities.

A record consisted of a stag beetle observation at a given place and time, independently of the number of individuals observed [41]. Geographical coordinates of each record were obtained with a precision of 10 × 10 km, to produce a distribution map of this species using the Universal Transverse Mercator (UTM) grid. Records for which 10 × 10 km coordinates could not be obtained were excluded from this study. Only observations up to 2019 were included. Undated records have been included but used only in quantitative analysis.

### 2.2. Data Analysis

For each data source, 10 × 10 km UTM grid cells (hereafter, grid cells) were sorted into two groups: (1) unique grid cells, i.e., grid cells contributed exclusively by one source, and (2) shared grid cells, i.e., grid cells contributed by more than one source.

We built two accumulated plots of new grid cells with time: (1) new grid cells, both unique and shared, contributed by each source, and (2) new grid cells contributed by each source, after removing those grid cells already contributed by any of the other sources. The first plot provided information about the contribution of each source to the total pool of grid cells, while the second plot provided information about the originality, or redundancy, of this contribution. Ties between sources, i.e., grid cells contributed for the first time in the same year by two sources, were scored for each source (n = 10 for publications, 8 for museums and 8 for citizen science).

The number of occupied grid cells per province were sorted by source, based on the source that first contributed each grid cell. Ties between sources were scored as a separate group. We produced three maps: percentage of grid cells contributed by each source at the provincial level, percentage of grid cells contributed by citizen science per province, and a distribution map of the European stag beetle in Spain, indicating those grid cells that have not yet been provided by citizen science.

GIS analysis and mapping were performed with ARCGIS 10.6 [42]. Maps were constructed in ETRS89 reference system by using the layers “Terrestrial 10 × 10 km grid” from Ministerio para la Transición Ecológica y el Reto Demográfico (*miteco.gob.es*) and “Líneas límite provinciales” from Instituto Geográfico Nacional (*centrodedescargas.cnig.es*).

All average values are given with their standard deviations.

## 3. Results

The database included 4167 records from 1784 to 2019, from which 3937 (132 undated) were georeferenced and were included in the following calculations. Citizen science provided 2452 records, museums 804 and publications 681. Overall, these records referred to 734 grid cells. Over 62% of the grid cells were unique and citizen science contributed more than 63% of these unique grid cells (Figure 1). Shared grid cells between sources ranged from 22.8 to 27.7%. Less than 11% of the grid cells were shared among the three sources.

Accumulation plots of grid cells with time showed three main patterns (Figure 2): (1) publications contributed the oldest grid cells, followed by museum and by citizen science; (2) the pace of accumulation of grid cells had a strong increase in the late 1970s and early 1980s; (3) both the number of grid cells and new grid cells contributed by publications and museums have increased at a slower pace after 2000.

Although citizen science contributed, on average, almost 50% of new grid cells per province, the contribution of each source differed among the 34 provinces occupied by the stag beetle (Figure 3A). Publications were the main source of grid cells for three provinces, museum for eight and citizen science for 19 (Figure 3A). In the remaining four provinces, a tie occurred between sources. There was no significant differences in the percentage of grid cells from museums and the presence or absence of museums in that province (36.8 ± 19.6, n = 19 vs. 26.5 ± 26.8, n = 15, F_1,32_ = 1.564, *p* = 0.220).

Citizen science accounted for 68.5 ± 22.4% (n = 34) of the grid cells reported per province. Citizen science accounted for 100% of the grid cells reported only in four provinces, while it accounted for less than 70% of the grid cells for 18 out of the 34 provinces with presence of stag beetle (Figure 3B). Grid cells not contributed by citizen science were scattered across the Spanish distribution of the stag beetle (Figure 4).

## 4. Discussion

We found that citizen science has made the largest contribution in delineating the distribution of the European stag beetle in Spain. Nevertheless, publications and museum data contributed uniquely almost 25% of all grid cells, as well as one third of the unique grid cells occupied by the stag beetle, and cannot be dismissed as useful sources of geographical distribution. These three sources of geographical information were complementary and geographic mismatches among sources may indicate differential biases. We discuss the implications of these results below.

Citizen science contributed the highest percentage of grid cells to the Spanish distribution of the stag beetle, despite its recent implementation compared to publications and museum data. Similar trends have been reported for studies addressing the Linnean shortfall [31]. For insects easily recognized, knowledge on distribution is being greatly improved thanks to citizen science-based studies [29,43,44,45], including the stag beetle [28,36,39,40]. The main advantage of citizen science was the large number of participants and its immediacy [30]. A potential drawback is the bias towards urban habitats [28]. The extent of such bias is difficult to assess in the Spanish case, with a high recruitment of volunteers from rural areas.

Museum data contributed a substantial proportion of unique grid cells, both at the national and provincial levels. The role of natural history collections in documenting taxonomical, historical and geographical dimensions of diversity has been emphasized [16,18]. This data set included the two main natural history museums in Spain, located in Madrid and Barcelona. In addition, it included 26 smaller collections, mainly from universities, 21 of which contributed unique grid cells. Entomological collections at Spanish universities are mainly nurtured through mandatory assignments to students in entomology courses [46]. Due to the recruitment of students from provinces nearby, these university collections host a representation of the biota beyond narrow provincial limits [46]. This could explain the lack of relationship between the presence of museum collections and the contribution of museums to unique grid cells at the provincial level. Thus, the importance of provincial museums for geographical documentation of biodiversity should not be neglected [19]. In particular, appropriate curation and accessibility should be guaranteed [20], as well as sustained growth of their collections [47] by encouraging the fundamental contribution of students while ensuring the conservation of threatened species.

Publications also contributed a fair share of unique grid cells. This contribution highlights the importance of entomological publications at national and provincial levels. Despite a crisis in the publication of national scientific journals [15], at least ten entomological journals are published in Spain [48]. Therefore, it is of strategic interest to keep a healthy national and local entomological community, which provides an often underestimated contribution in filling the Wallacean shortfall [32,49]. An active entomological community provides local expertise and sustains a network of national entomological journals. It is also priority that these journals keep encouraging the publication of a broad range of faunistic records, beyond the usual emphasis on rare species or novelties at national or provincial levels.

The high proportion of grid cells contributed uniquely by museums and publications indicated that 15 years of citizen science were unable to yield a complete view of the distribution of the stag beetle in Spain. In other studies, citizen science has provided accurate distribution information in a very short time [29]. In the case of the European stag beetle, two mapping projects involving short term (1–2 years) citizen science found only slight mismatches with respect to historical distribution data obtained by a combination of museum data, published records and expert data [28,36]. Two factors can influence the success of these attempts: (1) the number of occupied grid cells was less than half of the occupied grid cells in Spain (ca. 200 in UK [28]; 288 in Italy [50]) and (2) the distribution of the stag beetle in UK and Italy is within highly populated areas, compared to Spain (*ec.europa.eu/eurostat/*). In the face of a broad distribution of the stag beetle and the low density of recorders in many provinces, a longer term approach can be the only option. A long term approach and annual calls for participation are also issued in France [35], where the stag beetle has a broad distribution. The main challenge in long term mapping efforts through citizen science is to keep motivation [30]. In the Spanish case, as in other programs [36], the key to sustained support has been personalized feedback to each participant, although citizen contribution has been mostly opportunistic (sensu [51]).

The long term approach to distribution mapping of the stag beetle in Spain does not allow a straightforward comparison of trends in occupancy, by contrast with the repeated monitoring using short-term campaigns [28,36,39,40]. Thus, it is difficult to ascertain the reasons of the mismatch between citizen science data and historical data provided by museums and publications. Eventually, long-term citizen science is expected to hit on the grid cells previously reported uniquely by museums and publications, excepting for a few remote locations or some true local extinctions. Nevertheless, the provincial data presented here suggest that the extent to which citizen science can overtake the role of museums and publications is dependent on geographical biases by all the data sources [31].

Given the pros and cons of museum data, publications and citizen science, we advise their combined use in cases, such as the European stag beetle in Spain, in which a broad, heterogeneous, sparsely populated territory has to be prospected. We foresee three possibilities of collaboration, not competition, among data sources. First, citizen science contributed to online repositories has led to publications [52]. Second, highly motivated volunteers have been encouraged to report their records through publications rather than through citizen science [53]. Third, volunteers can contribute not only with field records but also with data retrieval from entomological collections, whose repeated survey is logistically complicated for professional researchers. Given that data retrieval from museums is difficult [31], this collaboration in surveying museum data should be encouraged. Beyond their value as sources of mapping data, we should not dismiss museum data and amateur entomological publications given their irreplaceable role as sources of historical information, that is fundamental for conservation purposes [14,31,54,55]. Finally, museum data are the only insurance against information decay due to changes in taxonomic status [56].

In conclusion, citizen science has played a fundamental role in delineating the Spanish distribution of an emblematic, threatened beetle, and it is expected to increase its contribution in the future. However, museum data and publications should not be neglected as important sources of information, both due to geographical biases in the contribution of citizen scientists and as irreplaceable sources of historical data.

## Figures and Tables

**Figure 1 insects-12-00202-f001:**
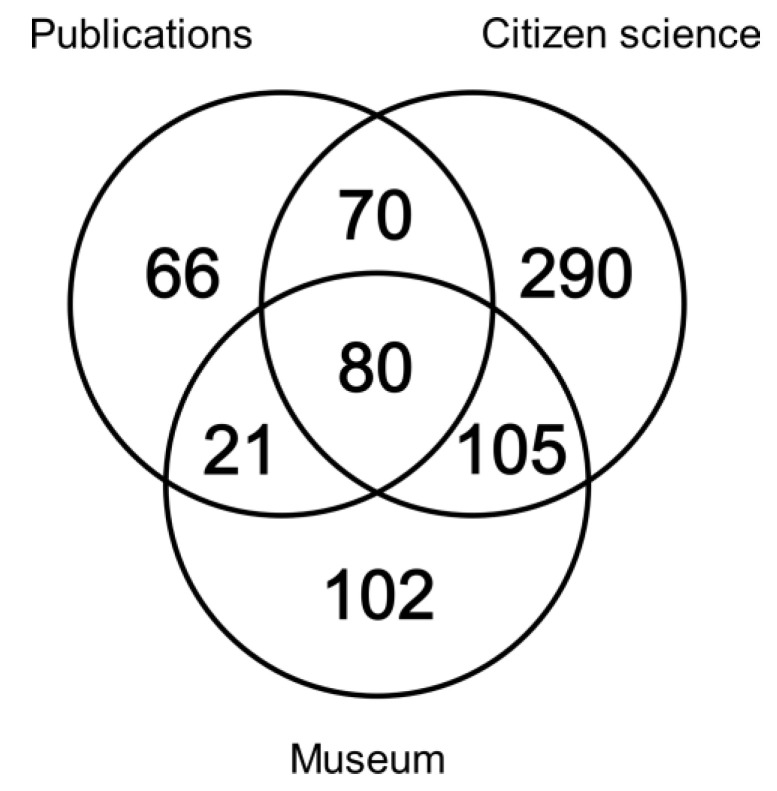
Contribution of unique and shared 10 × 10 km Universal Transverse Mercator (UTM) grid cells by each data source to the distribution map of the European stag beetle in Spain.

**Figure 2 insects-12-00202-f002:**
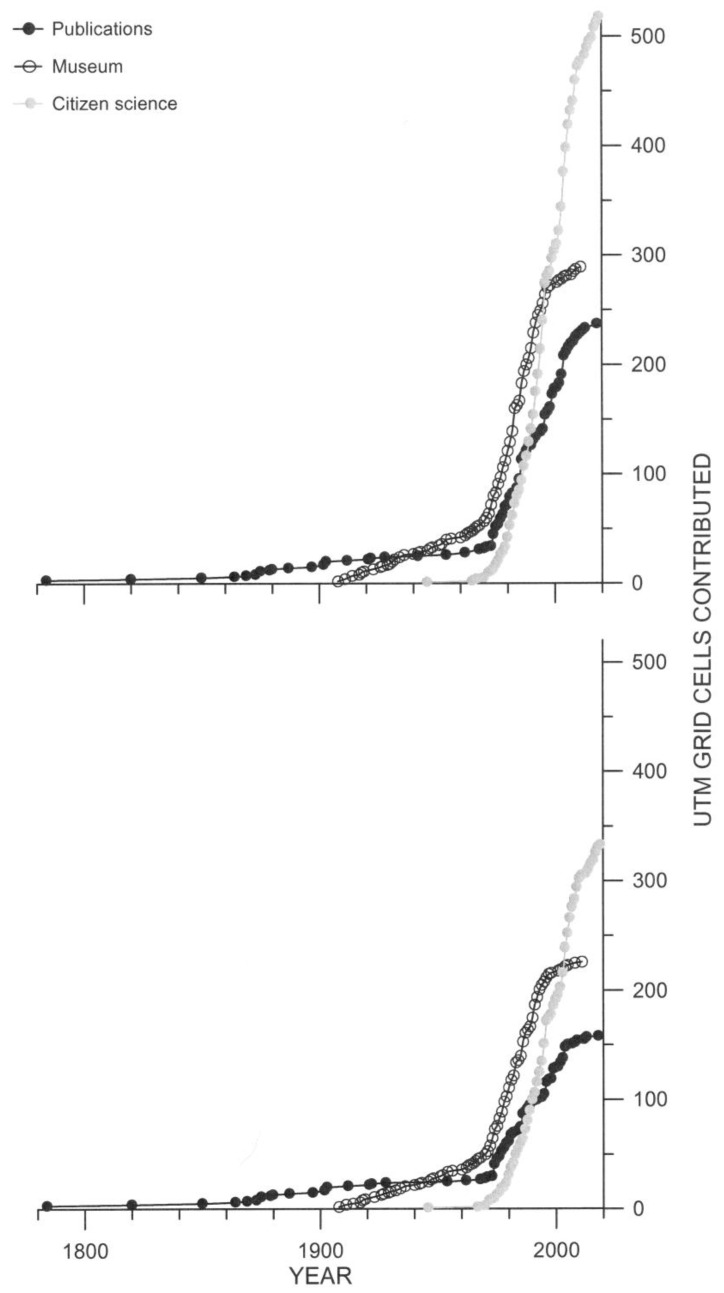
Accumulation with time of grid cells contributed by three sources of records of the European stag beetle in Spain. Upper panel, total grid cells (both unique and shared); lower panel, new grid cells.

**Figure 3 insects-12-00202-f003:**
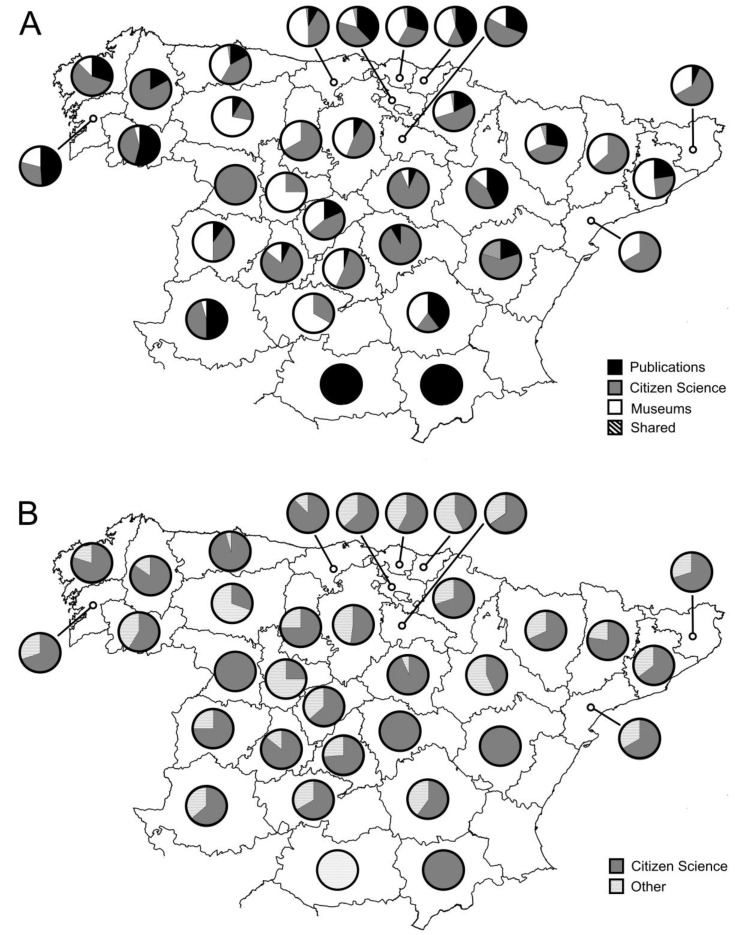
(**A**). Relative contribution by each data source to the number of grid cells known per province. (**B**). Relative number of grid cells contributed by citizen science, including those shared with other data sources, and grid cells uniquely contributed by other data sources.

**Figure 4 insects-12-00202-f004:**
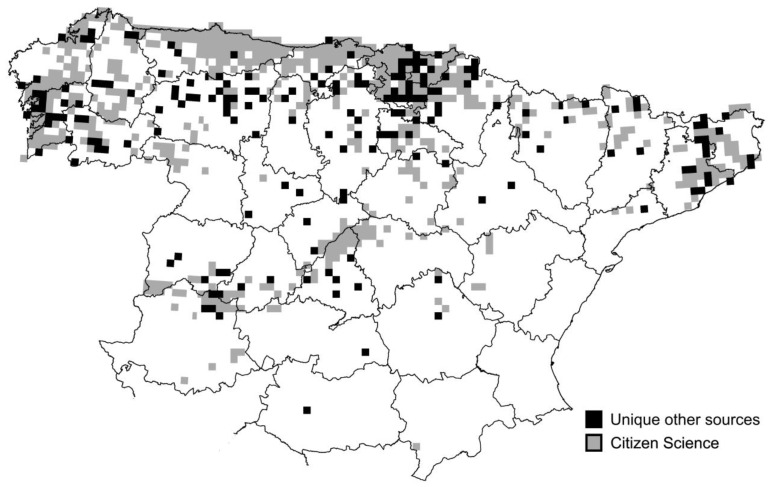
Distribution of the European stag beetle in Spain. Grey grid cells have been contributed by citizen science while black grid cells have been uniquely contributed by other data sources (publications or museum data).

## Data Availability

Data are available upon request to the first author.

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
