# Peer review of "Relative Contribution of Citizen Science, Museum Data and Publications in Delineating the Distribution of the Stag Beetle in Spain"

_insects, 2021, doi:10.3390/insects12030202_

Round 1

Reviewer 1 Report

The article is well done and addresses the topic in an interesting and rational way.

Only a few comments and observations follow.

Line 89 – I think there em-dashes without spaces should be used

Line 143 – Please use “database” and not “data base” as in the M&M.

Line 192 – “Knowledge on insect distribution is being greatly improved thanks to citizen science-based studies”. However, CS can only work with a few easly recognizable insects (not for example with those too small in size). Please underline this point.

Line 254 – “Third, volunteers can contribute not only field records but 255 also retrieve data from entomological collections”. I suggest to modify as: Third, volunteers can contribute not only with field records but also with data retrieval from entomological collections.

Author Response

Reviewer 1

Line 89 – I think there em-dashes without spaces should be used

Answer: done.

Line 143 – Please use “database” and not “data base” as in the M&M.

Answer: done.

Line 192 – “Knowledge on insect distribution is being greatly improved thanks to citizen science-based studies”. However, CS can only work with a few easly recognizable insects (not for example with those too small in size). Please underline this point.

Answer: this important point has now been included.

Line 254 – “Third, volunteers can contribute not only field records but 255 also retrieve data from entomological collections”. I suggest to modify as: Third, volunteers can contribute not only with field records but also with data retrieval from entomological collections.

Answer: reworded as indicated.

Reviewer 2 Report

Dear authors,

Please see the attachment. I have provided several comments and suggestions that I hope you will find useful and constructive.

Author Response

Reviewer 2

Line 8 - Conservation of threatened species in general, not just insects.

Answer: of course. We refer here to insects because this journal is addressing an specific audience, as the title of the journal indicates.

Line 14 - You mean grid cells?

Answer: yes, sorry. We have reworded "squares" to "grid cells" throughout the manuscript.

Line 17 - territory

Answer: changed.

Line 18 - See my first comment

Answer: we agree that distribution maps are in the basis of conservation of all species, but we consider appropriate to focus our message to the entomological audience. A more general statement seems more appropriate for a broader audience such as readers of generalist journals like Biological Conservation or Diversity and Distributions.

Line 25 - See my second comment

Answer: modified to "grid cells" throughout the manuscript.

Line 28 - This needs rephrasing. It would be naive and anti-scientific to even suggest that museum collections/specimens and scientific publications do not have any value in addressing the Wallacean shortfall and its consequences. Citizen science can be really helpful in filling some gaps, but it must be approached with caution, since the occurrence data need taxonomical validation by experts. Otherwise, this type of occurrences might lead to substantial biases and errors.

Answer: we totally agree and this was not absolutely the message we want to convey; rather the opposite. We have simply removed this comment.

Line 31 - heterogeneous

Answer: corrected.

Line 41 - The Wallacean shortfall refers not only to species of conservation concern, but to species in general.

Answer: rephrased to convey the wider meaning of Wallacean shortfall.

Line 49 - What do you mean by this? That macroecological analyses and publications are favored against taxonomical research, since macroecological journals have higher IF?

Answer: Not necessarily macroecological; simply, that faunistic publications are not considered a scientific output at the same level of academic excellence than phylogenies, community ecology, conservation ecology, conservation genetics, ecophysiology, pest control or many other disciplines which entomologists can potentially perform. We have rephrased this sentence.

Line 52 - That has nothing to do with conservation or threatened species.

Answer: rephrased.

Line 56 - GBIF does not seem to have a geographical bias - at least for several organisms.

Answer: we did not have GBIF in mind, but museums, which are physical public repositories of biodiversity data. Each museum is likely to have a geographical bias, except may be very large, prestigious museums, able to get specimens from all over the world. We have rephrased this sentence to avoid confusion.

Line 62 - What about lack of taxonomical expertise and thus taxonomical uncertainty?

Answer: added.

Line 83 - grid cells?

Answer: yes; corrected throughout the manuscript.

Line 101 - At least part of the museum data could have been retrieved from GBIF. Did you query the GBIF database for your species? It hosts 1721 species occurrences from Spain, 10% of which come from iNaturalist, the biggest citizen-science online platform.

Answer: the database we have compiled over 15 years predates the existence of GBIF. Actually, many data currently present in GBIF about the European stag beetle in Spain, both museum data and direct observations, have been contributed by our effort, through the Spanish Ministry of Agriculture to whom periodical reports are sent, because they have been using our database to meet the mandate of the EU of monitoring the status of the stag beetle (we have added this information to the Methods section). Thus, quering GBIF in this particular case mostly yields our own, hard-gained information. Recent additions to GBIF from iNaturalist were deliberately excluded, as well as data from other citizen science platforms such as observado.com and biodiversidad virtual, because they need careful scrutiny of each record. We are aware of certain curation of the data included in these platforms, but we are also aware of inclusion of data based on blurred pictures in which diagnostic traits are not visible. Given recent reports of Lucanus tetraodon and L. pontbrianti in Spain, we took a conservative approach and we excluded any data, including museum data, that were not directly authenticated by us or trained colleagues.

We intend to perform a comparison between our database and the ones available in Biodiversidad Virtual and observado.com, which will serve to compare the efficiency of our approach with that of virtual platforms of citizen science. However, this is beyond the objectives of the present study.

Line 109 - Which project?

Answer: actually, it is a working group of Lucanidae included in the Sociedad Entomológica Aragonesa (Aragonian Entomological Society). This working group has now been replaced the more cryptic "project" mentioned previously.

Line 112 - Do you mean iNaturalist here or something else? It's not clear whether you queried iNaturalist or not.

Answer: we did not query iNaturalist. As explained above, this would require detailed scrutiny of each record and we lack at the moment manpower to perform such task. We only included a few data from small national platforms in which we could doublecheck the accuracy of the identifications. Virtual platforms are growing exponentially and checking the accuracy of each record is becoming a daunting task. As explained above, we intend to enter that area in the near future (we are about to get a technician that will be trained for accurate identification of the four Spanish Lucanus species) but this has not been possible until now with the workforce at hand. We have now added a short comment indicating that large virtual platforms were not included in our database.

Line 116 - You did not have even coarse geographical information for these data? Did you follow any georeferencing protocol to eliminate or significantly reduce this kind of records? A grid cell of 100 km2 is pretty big, so it seems odd that at least a number of records could not be attributed to any cell. Even old museum specimens provide a description of the locality the specimen was collected and more often than not, we are able to at least roughly approximate the collection locality.

Answer: in some cases, it was impossible to locate the place (name ambiguous that could correspond to several, far apart, towns; only province indicated; misspelled name in the label, or using a shortened spelling that could correspond to several places... the casuistic is large). Anyway, the proportion of records without georeference was only 5% of the total available. You can be sure we have spent lot of time trying to georeference every possible record and little room for improvement remains.

Line 117 - You need to present these values in the Results section (how many are the total records, the number of georeferenced records, those that were excluded for various reasons, etc).

Answer: this information has been added at the beginning of the Results section.

Line 120 - grid cells

Answer: corrected throughout the manuscript.

Line 144 - out of how many?

Answer: 734 is the number of currently known grid cells occupied by the European stag beetle in Spain. Thus, they are 100%. We believe that indicating the number of total grid cells of Spain is not informative.

Line 156 - This figure is very pixelated. Please provide a better quality (higher resolution) figure.

Answer: we provide a figure with higher resolution and we have modified the layout and the size of the dots to improve readability.

Line 174 - What does the grey color depict in the inset? The provinces for which occurrence data are available?

Answer: yes. Anyway, we realized that this inset was not needed and we have removed it from the new version of Fig. 3.

Line 175 - This figure needs to be enlarged and provided in color. I can hardly read it as it is.

Answer: we have redrawn the figure by combining the A and B panels in a single figure, and with higher contrast, so we believe the figure should be readable with no need of adding colour.

Line 178 - You mean other sources, right? Because if you excluded other sources, you are left again with citizen science sources. This figure needs to be in color, as I can hardly discern the light grey cells from the white background. Are these the 10 x 10 grid cells you were talking about? So for a very large fraction of Spain, you do not know if the species exists there, right? Most of the collections seem to be in northern Spain. Did you run any sampling bias check? You can do this with the sampbias R package. It would be a great addition to your manuscript and provide further insights regarding the concentration of collection effort either by professionals or amateurs in Spain.

Zizka, A., Antonelli, A., & Silvestro, D. (2021). sampbias, a method for quantifying geographic sampling biases in species distribution data. Ecography, 44(1), 25-32.

Answer: sorry for the confusion. "Excl." was intended to mean "exclusively", not "excluded". We have reworded the legend to clearly indicate "Unique other sources". This means that black grid cells were uniquely contributed by other sources (museum and publications), not by citizen science.

For the large empty portions of Spain we are pretty certain that the species does not occur. This is an iconic species, highly appreciated by entomologists. It has been searched for in every single province of Spain, including all the South and East, were has never been found. If not reported already, we are certain it is not present. We have unpublished support from habitat models, which indicate that we have covered well the suitable areas. Any remaining gaps in the distribution are from areas in which this species is already known. Thus, only "fine work" and gaps in prospected areas remain to be filled. We are not at a very early stage of knowledge of the distribution of this species in Spain; this was the situation in 1994, when we started this database. We believe the bias check is not needed. We have added a comment on the current coverage of this distribution map in the Methods section.

Line 179 - You should clarify whether or not you used GBIF data. If not, you should and you should state how many occurrences come from iNaturalist. Then, you can use the coordinatecleaner R package to run some diagnostic tests regarding the quality of these occurrences and report them as well. This way your manuscript will be greatly improved.

Zizka, A., Silvestro, D., Andermann, T., Azevedo, J., Duarte Ritter, C., Edler, D., ... & Antonelli, A. (2019). CoordinateCleaner: Standardized cleaning of occurrence records from biological collection databases. Methods in Ecology and Evolution, 10(5), 744-751.

Answer: as explained above, we have contributed all our database to GBIF, not the other way around. iNaturalist was not included in the database for the reasons stated above. We honestly believe our database is of high quality and with contrasted accuracy and including more data in a rush, with no time to check the accuracy of the identifications, would decrease the quality of the information. As stated above, we already plan to perform a comparison between the coverage of our database compared to other major repositories used in Spain (observado.org and Biodiversidad Virtual) but this is next step, not the aim of this study.

Line 195 - Not it's not, if you follow my suggestion about sampbias and coordinatecleaner R packages.

Answer: We suppose this is straightforward if records are utilised in this bias analysis. We are less certain that this approach can be directly used for grid cells of 10 x 10 km because it is not possible to score unambiguously each grid cell in our database to rural or urban areas.

Line 215 - it

Answer: corrected.

Line 232 - This site does not provide information regarding the overlap you mention.

Answer: sorry, this webpage shows the population density in Europe and each reader needs to overlap this map with the distribution map of the stag beetle. We have rephrased this sentence to avoid confusion.

Reviewer 3 Report

Manuscript ID: insects-1115255: Relative role of citizen science, museum data and publications in delineating the distribution of the stag beetle in Spain.

General comment by Reviewer

The manuscript presented by the authors provides an integrated basis for deepening the knowledge of particularly threatened species. However, the results seem almost obvious given the citizen science projects in progress and already widely discussed in terms of results compared to the classic forms of monitoring. No details are given on environmental quality, which is very important for these species.

In any case, the overall judgment is positive for such a vast and relatively little explored area.

I provide several comments in the text.

Reviewer TITLE: why is the role relative?

Reviewer L71: From the general to the particular too clearly. It is necessary to combine the concepts with respect to the species studied.

Reviewer L97: The first author ?, this should be moved to Author contributions

Reviewer L238: ... and in Italy? LIFE MIPP, INNAT

Author Response

Reviewer 3

The manuscript presented by the authors provides an integrated basis for deepening the knowledge of particularly threatened species. However, the results seem almost obvious given the citizen science projects in progress and already widely discussed in terms of results compared to the classic forms of monitoring.

Answer: we agree that citizen science is proving a reliable source of distribution data for easily recognizable insects. An increasing number of articles is reporting successful monitoring projects and successful enrolment of citizens in monitoring of insects. In this context, successful means (1) reliable data and (2) many records. However, we were not able to find but two studies (cited in our manuscript) that quantified the contribution of citizen science data, compared to classic monitoring. We believe our study provides (1) novel information by performing a quantitative comparison of citizen science, museum data and publications and (2) a cautionary tale by showing a large number of grid cells contributed by museum data and publications even after 15 years of engagement of citizens.

No details are given on environmental quality, which is very important for these species.

Answer: we are not sure we understand this comment. MM has performed environmental niche modelling of this beetle in the Iberian Peninsula using several analytical approaches. These yet unpublished results show that current knowledge of the Iberian distribution of the stag beetle is reliably captured by current records. Stag beetle does not remain to be reported from major parts of the Spanish territory; its limits have been reliably established and the gaps remaining are within provinces in which the species is already known to occur. New grid cells are reported every year, but always in the proximity of already occupied grid cells. In this sense, we are certain that major empty zones in Spain, in between the main nucleus in N Spain and the populations in the mountains of central Spain, are not due to lack of reporting but to unsuitable environmental conditions, namely, too warm areas. We have included a short statement in the Methods to clarify the coverage of this database.

In any case, the overall judgment is positive for such a vast and relatively little explored area.

Answer: thanks.

Reviewer TITLE: why is the role relative?

Answer: we mean the relative contribution of citizen science, compared to museum data and publications. We have slightly rephrased the title to improve clarity.

Reviewer L71: From the general to the particular too clearly. It is necessary to combine the concepts with respect to the species studied.

We honestly do not understand with correction we are expected to perform here.

Reviewer L97: The first author?, this should be moved to Author contributions

Answer: MM is already indicated as responsible for data curation in the Author Contributions section. We have added MM as responsible of investigation and removed the mention in methods.

Reviewer L238: ... and in Italy? LIFE MIPP, INNAT

Answer: our comment in this sentence intended to explicitly acknowledge our personalized feedback in the Spanish case. This was possible because no platform was used to get the records. Records were directly sent to MM, who immediately and personally answered all contributors, indicating the degree of novelty of each record. We have now added that feedback was also given in the LIFE MIPP case, but we did not find published information about INNAT.

Round 2

Reviewer 2 Report

Dear authors,

Thank you for your detailed responses to all the issues raised. I have no further comments to make.